# Brain Tissue Oxygenation-Guided Therapy and Outcome in Traumatic Brain Injury: A Single-Center Matched Cohort Study

**DOI:** 10.3390/brainsci12070887

**Published:** 2022-07-06

**Authors:** Sami Barrit, Mejdeddine Al Barajraji, Salim El Hadweh, Olivier Dewitte, Nathan Torcida, Joachim Andre, Fabio Silvio Taccone, Sophie Schuind, Elisa Gouvêa Bogossian

**Affiliations:** 1Department of Neurosurgery, Hopital Erasme, Université Libre de Bruxelles (ULB), 1070 Brussels, Belgium; samibarrit@gmail.com (S.B.); mejdi.albarajraji@gmail.com (M.A.B.); salimhadweh15@gmail.com (S.E.H.); olivier.de.witte@erasme.ulb.ac.be (O.D.); sophie.schuind@erasme.ulb.ac.be (S.S.); 2Department of Neurology, Hopital Erasme, Université Libre de Bruxelles (ULB), 1070 Brussels, Belgium; nathan.torcida.sedano@erasme.ulb.ac.be; 3Department of Radiology, Hopital Erasme, Université Libre de Bruxelles (ULB), 1070 Brussels, Belgium; joachim.andre@ulb.be; 4Department of Intensive Care, Hopital Erasme, Université Libre de Bruxelles (ULB), 1070 Brussels, Belgium; fabio.taccone@ulb.be

**Keywords:** brain oxygenation, outcome, protocolized care, head injury, trauma

## Abstract

Brain tissue oxygenation (PbtO_2_)-guided therapy can improve the neurological outcome of traumatic brain injury (TBI) patients. With several Phase-III ongoing studies, most of the existing evidence is based on before-after cohort studies and a phase-II randomized trial. The aim of this study was to assess the effectiveness of PbtO_2_-guided therapy in a single-center cohort. We performed a retrospective analysis of consecutive severe TBI patients admitted to our center who received either intracranial pressure (ICP) guided therapy (from January 2012 to February 2016) or ICP/PbtO_2_-guided therapy (February 2017 to December 2019). A genetic matching was performed based on covariates including demographics, comorbidities, and severity scores on admission. Intracranial hypertension (IH) was defined as ICP > 20 mmHg for at least 5 min. Brain hypoxia (BH) was defined as PbtO_2_ < 20 mmHg for at least 10 min. IH and BH were targeted by specific interventions. Mann–Whitney U and Fisher’s exact tests were used to assess differences between groups. A total of 35 patients were matched in both groups: significant differences in the occurrence of IH (ICP 85.7% vs. ICP/PbtO_2_ 45.7%, *p* < 0.01), ICU length of stay [6 (3–13) vs. 16 (9–25) days, *p* < 0.01] and Glasgow Coma Scale at ICU discharge [10 (5–14) vs. 13 (11–15), *p* = 0.036] were found. No significant differences in ICU mortality and Glasgow Outcome Scales at 3 months were observed. This study suggests that the role of ICP/PbtO_2_-guided therapy should await further confirmation in well-conducted large phase III studies.

## 1. Introduction

Traumatic brain injury (TBI) is a major public health burden, causing death and disability worldwide [1]. The global incidence of TBI is estimated to be around 27 to 69 millions of events per year [1,2]; in order to potentially improve long-term outcomes of these patients, prevention and treatment of secondary brain injury is essential since, secondary brain injury is a major determinant of outcome after TBI [3]. Intracranial pressure (ICP) and cerebral perfusion pressure (CPP) monitoring are widely used to optimize cerebral hemodynamics after severe TBI, since intracranial hypertension is associated with worse outcome in this setting [4]. However, no high-quality evidence suggests that this approach may improve neurological recovery in TBI patients [5]. Moreover, brain hypoxia, that might exist even in the absence of intracranial hypertension, i.e., because of microvascular dysfunction, systemic hypotension, hypoxemia, fever or anemia, is also an important cause of secondary brain injury and independent predictor of poor outcome [6,7,8,9].

As such, brain tissue oxygen (PbtO_2_) monitoring allows the detection of brain hypoxia in acute brain injured patients [6,10]. Although three ongoing randomized clinical trials investigating the effect of PbtO_2_-guided therapy on the outcome of TBI patients, the current evidence suggesting that PbtO_2_-guided therapy provides beneficial effects on survival and neurological recovery is based only one phase-II randomized study [11] and several retrospective or before-after cohort studies, in which the populations were significantly unbalanced at baseline [12,13,14,15,16,17]. Moreover, some cohort studies have failed to show any significant benefit of PbtO_2_-guided therapy on neurological outcome of these patients [18,19,20,21,22].

Therefore, the aim of this study was to assess the effectiveness of ICP/PbtO_2_-guided therapy compared to ICP-guided therapy on neurological outcome and global management of TBI patients.

## 2. Materials and Methods

We conducted a single-center retrospective cohort study of severe TBI patients admitted to the intensive care unit (ICU) of Erasme Hospital (Brussels, Belgium) from January 2012 to December 2019. All adult (>18 years) patients admitted with severe TBI (Glasgow coma scale, GCS < 9) during the study period were eligible for inclusion, if they needed an ICP monitoring in the first 48 h after admission. The decision to further monitor patients with PbtO_2_ probes was determined by device availability, which became standard of care since February 2017. The sole exclusion criterion was imminent death, resulting into early limitation of life-sustaining therapies. This study was approved by the Erasme Hospital (Université Libre de Bruxelles, Brussels, Belgium) ethics committee (2021/099), which waived the need for informed consent. This study was performed in accordance with relevant scientific and ethical guidelines and regulations. We obtained data through the electronic chart system used at Erasme Hospital.

### 2.1. Data Collection

We collected demographic data and presence of comorbidities. Clinical severity scores on admission, such as the Sequential Organ Failure Assessment (SOFA) and the Acute Physiology and Chronic Health Evaluation (APACHE) II scores were recorded [23]. The Marshall score was used to classify patients’ severity according to the initial computerized tomography scan [24]. The basic therapy intensity level (TIL) score was used to assess the intensity of ICP treatment [25]. The GCS was assessed on admission and at ICU discharge. Patients who died during ICU stay were considered as GCS of 3 at discharge.

We also recorded interventions received by patients during the ICU stay (such as mechanical ventilation, vasopressor and inotropic support and kidney replacement therapy) and the development of complications, including seizures (i.e., convulsive or non-convulsive), intracranial hypertension and brain tissue hypoxia. We also recorded the therapeutic interventions used to treat intracranial hypertension and/or tissue hypoxia. We recorded ICU and hospital mortality, the Glasgow Outcome Scale (GOS) [26] at 3 months according to the report of the ICU-follow program and the occurrence of unfavorable neurological outcome (UO), defined as a 3-month GOS of 1–3, using medical reports from follow-up visits.

ICP and PbtO_2_ were continuously measured using intraparenchymal probes (Neurovent-P, Raumedic, Helmbrechts, Germany and IM3.ST_EU; Integra LifeSciences Corporation, Plainsboro, NJ, USA, respectively). Whenever possible, PbtO_2_ probes were placed into the hemisphere at greatest risk for secondary brain injury (i.e., close to the injured/contused area, through a frontal burr hole using a triple-lumen bolt). Probe location was confirmed by CT scan after placement. The adequate functioning of the probe was tested with a 100% oxygen fraction (FiO_2_) test for a maximum of 15 min_._

Intracranial hypertension was defined as at least one ICP value exceeding 20 mmHg for at least 5 min at any time. Brain tissue hypoxia was defined by a PbtO_2_ below 20 mmHg for at least 5 min [27]. All patients received tier 0 therapies [28], such as proper head positioning, CPP ≥ 60 mmHg, avoidance of neck compression, control of extra-cerebral cerebral injuries, prevention and treatment of fever and metabolic disturbances [29].

ICP-guided therapy included different therapeutic interventions (i.e., external ventricular drainage, hyperventilation, increased sedation, osmotic therapy, barbiturates or decompressive craniectomy) aiming to reducing ICP below 20 mmHg. PbtO_2_-guided therapy was considered as all specific therapeutic interventions (induced hypertension, red blood cells transfusions, changes in ventilatory settings and PaCO_2_) aiming to achieve a PbtO_2_ > 20 mmHg.

### 2.2. Study Outcomes

Primary outcome was the impact of ICP/PbtO_2_ guided therapy on neurological status at ICU discharge assessed by the GCS. Secondary outcomes included: (a) impact of ICP/PbtO_2_-guided therapy on the development of intracranial hypertension; (b) the impact of ICP/PbtO_2_ guided therapy on hospital mortality; (c) the impact of ICP/PbtO_2_ on neurological outcome at 3 months.

### 2.3. Statistical Analysis

Descriptive statistics were generated for all variables. Numeric variables were described as median and interquartile intervals 25–75% or mean and standard deviation. Categorical variables were described as numbers and proportions. Normally distributed continuous variables were compared using t Student test and asymmetrically distributed variables were analyzed using Mann-Whitney test. Categorical variables were compared using chi square or Fisher’s exact test. A genetic matching (a multivariate matching method using an evolutionary search algorithm based on heuristic rules to optimize covariate balance) [30] was performed based on demographics, comorbidities, and severity scores on admission (Figure 1). Statistics and matching were computed on the open-source R software, version 3.6.3 (R statistical and computing software; http://www.r-project.org/ (accessed on 15th October 2021)) using the MatchIt and Matching packages and the software SPSS 27.0 for MacIntosh. A sensitivity analysis of study outcomes was performed using the entire study cohort; adjusted odds ratio and confidence intervals based on variables on admission with a *p* value < 0.05 at univariate analysis were computed. A *p* value < 0.05 was considered statistically significant.

## 3. Results

Among 389 admitted patients over the study period, we identified 106 severe TBI patients that met the inclusion criteria. Among those, 35 (33%) were monitored with PbtO_2_ and received ICP/PbtO_2_-guided therapy, with a mean age of 51 ± 19 years and being predominantly male (59%). The most frequent comorbidities were alcohol use and arterial hypertension. Traumatic SAH was observed in 90% of patients and the presence of subdural/epidural hematomas were found in 53% of patients. The median GCS on admission was 5 (3–8). Intracranial hypertension was diagnosed in 73/106 (69%) patients and brain hypoxia in 24/35 (69%) patients. The characteristics of the study population are reported in Table 1. The distribution of patients according to monitoring strategy per year is illustrated in Appendix A.

### 3.1. Genetic Matched Cohort

Among the 71 patients in the ICP-guided therapy group, 35 patients were matched with the ICP/PbtO_2_-guided therapy cohort (Figure 1). The characteristics of the two groups are shown in Table 2; groups were well-balanced for most of the relevant variables. GCS at ICU discharge was higher, although not statistically significant, in the ICP/PbtO_2_ group when compared to the ICP group [10 (3–14) vs. 3 (3–12); *p* = 0.46]. The occurrence of intracranial hypertension was significantly lower in the ICP/PbtO_2_-guided than the ICP-guided therapy group (16/35, 46% vs. 27/35, 77%; *p* = 0.01); in particular, PbtO_2_-guided therapy was associated with a relative risk reduction of 41% [95% CIs 11–60%] and an absolute risk reduction of 31% [95% CIs 9–50%] in the risk of developing intracranial hypertension. The occurrence of hospital mortality (13/35, 37% vs. 18/35, 51%; *p* = 0.34) and UO (19/35, 54% vs. 25/35, 71%; *p* = 0.22) were numerically lower, although this difference was not statistically significant, in the ICP/PbtO_2_-guided therapy group when compared to the other (Figure 2).

### 3.2. Entire Cohort Analysis

Patients monitored with ICP/PbtO_2_ were younger and had higher APACHE and SOFA score on admission compared to others (Table 1). However, GCS on admission was similar between the two groups [5 (3–7) vs 5 (3–9), *p* = 0.72]. The intensity of therapy for intracranial hypertension (basic TIL score) was similar in both groups, although the ICP/PbtO_2_guided therapy group received more frequently vasopressors. The incidence of intracranial hypertension was lower in the ICP/PbtO_2_ group when compared to the other (16/35, 46% vs. 57/71, 80%; *p* = 0.001). In the multivariable model, ICP/PbtO_2_-guided therapy was associated with a lower chance of developing intracranial hypertension, when adjusted for Marshall score, GCS on admission and the presence of reactive pupils on admission (Table 3). However, ICP/PbtO_2_-guided therapy had no impact on mortality (unadjusted OR 0.51 [95% CI 0.22–1.18]) nor on neurological outcome (unadjusted OR 0.43 [95% CI 0.19–1.01]), even after adjustment for confounders such as intracranial hypertension, GCS on admission and age (adjusted OR 1.33 [95% CI 0.46–3.87] and adjusted OR 0.71 [95% CI 0.25–2.03], respectively—Appendix A). Appendix A) illustrates the rate of hospital mortality and neurological outcome at 3 months over the study period, according to monitoring strategy. The Kaplan-Meier curve of survival according to the monitoring strategies (ICP and ICP/ObtO_2_ group) was shown in Appendix A.

As a sensitivity analysis we also performed a comparison between ICP and ICP/PbtO_2_ in patients admitted after 2017, to reduce the possible bias of time in the previous analysis. The results are presented in Appendix A. Patients had similar GCS, APACHE, SOFA and Marshall scores on admission and had no statistical difference in outcomes (i.e., hospital mortality and GOS at 3 months). The incidence of intracranial hypertension remained lower in the ICP/PbtO_2_ group when compared to the other.

## 4. Discussion

In this matched cohort study, no significant differences in hospital mortality or the occurrence of UO at 3 months was observed between TBI patients undergoing ICP-guided and ICP/PbtO_2_ guided therapy. Interestingly, patients monitored with ICP/PbtO_2_ developed less intracranial hypertension than those monitored with ICP alone. These results were also confirmed when the entire study cohort was analyzed.

PbtO_2_ values derive from a complex balance between oxygen supply, demand, and extracellular diffusion [10,31]. In the management of TBI, PbtO_2_ and ICP/CPP are used as surrogate endpoints of potential pathological processes, such as mass effect, cerebral edema, tissue hypoxia or reduced cerebral blood flow. The relationship among all these biomarkers’ variables remains not entirely understood. For instance, brain hypoxia may occur despite ICP/CPP being within normal ranges, due to diffusion-limited oxygen delivery caused by endothelial swelling, microvascular collapse, and perivascular edema [31]. This supports the rationale behind PbtO_2_-guided therapy [10]. Moreover, PbtO_2_ is thought to provide unique information as an independent predictor of TBI outcome [32,33]. PbtO_2_-guided therapy impact may not be limited to counteract brain hypoxia but also offers an earlier and better preventive effect compared to sole-ICP guided therapy on other processes of secondary brain injury, such as cerebral edema, with a subsequent reduction on the occurrence of intracranial hypertension. 

Intracranial hypertension occurs frequently after TBI and is an important mechanism of brain injury that, if left untreated, can lead to death and poor outcome [34]. The Brain trauma foundation recommended starting treatment for intracranial hypertension when ICP exceeds the threshold of 22 mmHg [5]. However, the management of TBI patients should also include strategies for prevention of ICP surges, such as sedation and analgesia, elevation of the head, optimizing of head venous return, regardless of the ICP level [28]. Moreover, a target of 20–22 mmHg may not be adequate for all patients. The use of adjunctive monitoring such as PbtO_2_ [35] can help individualized ICP targets, leading to further reduction of ICP in the presence of brain hypoxia or a higher tolerance for slightly higher ICP if PbtO_2_ is adequate and the patients is waking up.

The decrease in the incidence of intracranial hypertension is interesting and may suggest an indirect favorable effect of PbtO_2_-guided therapy on ICP management in line with previous studies. In a randomized trial, Lin et al. [14] reported a significantly lower mean ICP and higher mean CPP when PbtO_2_ monitoring was implemented in the therapeutic algorithm of TBI patients. In the BOOST-II trial [11], consistent but not significant findings of reduced total intracranial hypertension burden were reported. Even in the retrospective studies where no effect of PbtO_2_-guided therapy on outcome was observed, PbtO_2_-guided therapy led to a trend towards lower levels of ICP and less episodes of brain hypoxia [18,20,22]. A meta-analysis by Xie et al. [13] also raised this putative effect of PbtO_2_-monitoring, while not being able to draw firm conclusions about it.

In our study, PbtO_2_-guided therapy had no impact on mortality or neurological outcome, as also reported in other studies [18,19,20,21,22]. However, the magnitude of the effect might be underestimated by the small number of analyzed patients in our study, as in previous cohorts. Ongoing large randomized trials [36,37,38] on PbtO_2_-guided therapy for TBI are sufficiently powered to detect significant changes in patient-oriented clinical outcomes (i.e., neurological outcome and survival) and might provide high quality data in the next years. Besides their highly anticipated functional and neurological outcomes, findings of ICP/CPP dynamics (i.e., mean or peak values as the total burden of intracranial hypertension) and management (i.e., targeted medical interventions, the use of decompressive craniectomy or other salvage therapies) would also be of great interest. Interestingly, patients monitored with ICP/PbtO_2_ presented longer ICU and hospital stays compared to the ICP group. This may be explained because patients in the ICP/PtbO_2_ group, despite similar severity on admission than the others, received more life support therapies, which prolonged survival time.

Due to its retrospective design, this study presents inherent limitations partially addressed by the genetic matching. No dynamic analysis of intracranial hypertension and brain tissue hypoxia episodes nor PbtO_2_- or ICP-guided interventions and their interrelations were performed. As a small single-center experience, the statistical power and generalizability of the findings are limited. Finally, potential confounding factors pertaining to neurocritical care evolution during the 8-years study time frame were not mitigated.

## 5. Conclusions

This single center matched cohort study was not able to detect any significant favorable effect on clinical outcomes of the implementation of PbtO_2_-guided therapy in severe TBI patients. However, our findings suggest that PbtO_2_-guided therapy may reduce the burden of intracranial hypertension in this setting.

## Figures and Tables

**Figure 1 brainsci-12-00887-f001:**
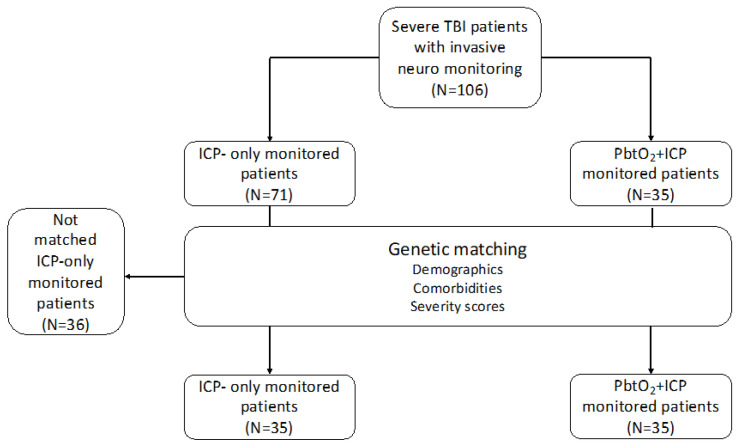
Flowchart of the genetic matching. TBI: traumatic brain injury; ICP: intracranial pressure; PbtO_2_: brain tissue oxygenation; ICU: intensive care unit.

**Figure 2 brainsci-12-00887-f002:**
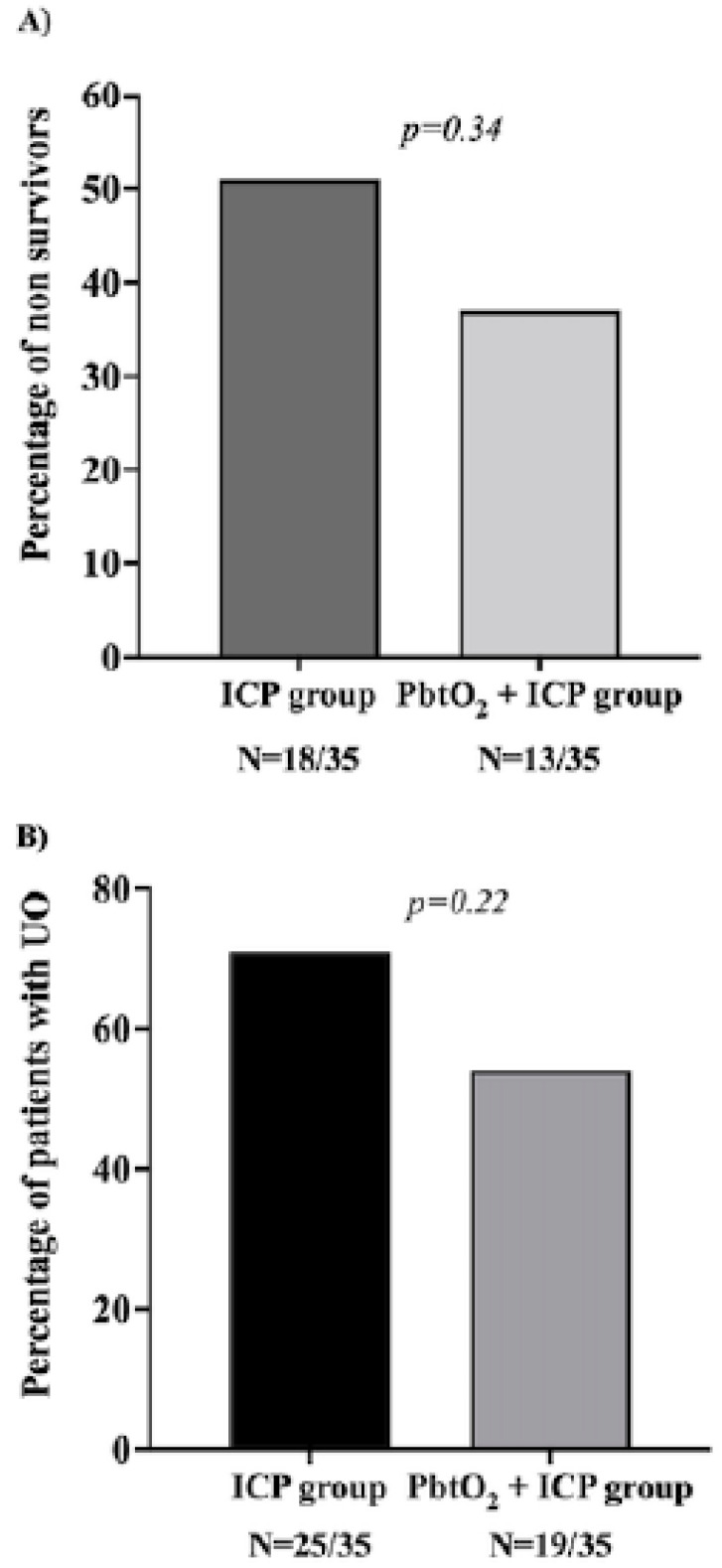
Outcomes according to type of monitoring strategy. Panel (**A**) hospital mortality; Panel (**B**) Unfavorable neurological outcome (UO) at 3 months (GOS 1–3). GOS: Glasgow outcome scale.

**Table 1 brainsci-12-00887-t001:** Characteristics of the study population. Data are presented as mean (±SD), count (%) or median (25th–75th percentiles).

	All Patients (N = 106)	ICP-Group (N = 71)	ICP/PbtO_2_-Group (N = 35)	*p*-Value
**Age, years**	51 (±19)	54 (±19)	45 (±17)	0.02
**Male gender, *n* (%)**	63 (59)	39 (55)	24 (69)	0.21
**Arterial hypertension, *n* (%)**	27 (26)	20 (28)	7 (20)	0.48
**Diabetes mellitus, *n* (%)**	7 (7)	5 (7)	2 (6)	0.99
**Heart disease, *n* (%)**	11 (10)	9 (13)	2 (6)	0.33
**Previous neurological disease, *n* (%)**	4 (4)	4 (6)	2 (6)	0.30
**Alcohol use, *n* (%)**	36 (34)	26 (37)	10 (29)	0.51
**Smoking, *n* (%)**	19 (18)	14 (20)	5 (14)	0.60
**COPD, *n* (%)**	3 (3)	3 (4)	0	0.55
**Liver cirrhosis, *n* (%)**	5 (5)	5 (6)	1 (3)	0.99
**Cancer, *n* (%)**	2 (2)	1 (1)	1 (3)	0.99
**Chronic kidney disease, *n* (%)**	3 (3)	3 (4)	0	0.55
**On admission**				
**APACHE score**	18 (15–21)	18 (15–21)	20 (17–23)	0.03
**SOFA score**	8 (4–10)	6 (4–10)	8 (8–10)	0.01
**GCS on admission**	5 (3–8)	5 (3–9)	5 (3–7)	0.72
**Marshall score, *n* (%)**				0.02
**1**	0	0	0	
**2**	5 (5)	3 (4)	2 (6)	
**3**	3 (3)	2 (3)	1 (3)	
**4**	27 (26)	25 (35)	2 (6)	
**5**	69 (65)	39 (55)	30 (86)	
**6**	2 (2)	2 (3)	0	
**Reacting pupils, *n* (%)**	79 (75)	53 (75)	26 (74)	0.99
**Traumatic SAH, *n* (%)**	95 (90)	67 (94)	28 (80)	0.04
**Epidural Hematoma, *n* (%)**	56 (53)	26 (37)	30 (86)	0.001
**Hypotension, *n* (%)**	40 (0.29)	24 (34)	16 (46)	0.29
**Hypoxemia, *n* (%)**	54 (51)	33 (47)	21 (60)	0.22
**Sodium, mmol/L**	138 (135–141)	138 (136–141)	137 (135–140)	0.19
**Glucose, mg/dL**	136 (124–170)	141 (126–176)	131 (122–167)	0.62
**Hemoglobin, g/dL**	12.1 (10.5–13.6)	11.9 (10.7–13.2)	12.2 (10.3–14.5)	0.63
**During ICU stay**				
**Mechanical ventilation, *n* (%)**	106 (100)	71 (100)	35 (100)	-
**Vasopressors, *n* (%)**	78 (74)	45 (63)	33 (94)	0.001
**Inotropes, *n* (%)**	10 (9)	4 (6)	6 (17)	0.08
**RRT, *n* (%)**	1 (1)	0	1 (3)	0.33
**EVD, *n* (%)**	93 (88)	66 (93)	27 (77)	0.55
**Complications**				
**Intracranial hypertension, *n* (%)**	73 (69)	57 (80)	16 (46)	0.001
**Brain tissue hypoxia, *n* (%)**	NA	NA	24 (69)	-
**Seizures, *n* (%)**	25 (24)	17 (24)	8 (23)	0.99
**Treatments**				
**TIL score basic**				0.13
**1**	20 (20)	11 (17)	9 (26)	
**2**	35 (35)	28 (43)	7 (20)	
**3**	16 (16)	10 (15)	6 (17)	
**4**	29 (29)	16 (25)	13 (37)	
**Osmotic therapy, *n* (%)**	61 (58)	43 (61)	18 (51)	0.41
**Hypothermia, *n* (%)**	23 (22)	15 (21)	8 (23)	0.99
**Barbiturates, *n* (%)**	19 (18)	15 (21)	4 (11)	0.29
**Decompressive craniectomy, *n* (%)**	25 (24)	14 (20)	11 (31)	0.23
**Outcomes**				
**ICU LOS, days**	9 (4–17)	7 (3–14)	16 (9–25)	0.001
**Hospital LOS, days**	17 (5–42)	10 (4–38)	30 (14–66)	0.006
**GCS at ICU discharge**	6 (3–13)	3 (3–12)	10 (3–14)	0.19
**ICU death, *n* (%)**	50 (47)	37 (52)	13 (37)	0.16
**Hospital death, *n* (%)**	51 (48)	38 (54)	13 (37)	0.15
**GOS at 3 months**	2 (1–4)	1 (1–4)	3 (1–4)	0.15

COPD = chronic obstructive pulmonary disease; APACHE: acute physiology and chronic health examination; SOFA: sequential organ failure assessment; GCS: Glasgow coma scale; GOS: Glasgow outcome scale; ICU: intensive care unit; LOS: length of stay; TIL: therapy intensity level; RRT: renal replacement therapy; EVD: external ventricular drainage; SAH: subarachnoid hemorrhage.

**Table 2 brainsci-12-00887-t002:** Comparison of the study matched cohort, according to the use of intracranial pressure (ICP) or ICP and brain oxygen pressure (PbtO_2_) monitoring. Data are presented as count (%) or median (25th–75th percentiles).

	ICP-Group N = 35	ICP/PbtO_2_ Group N = 35	SMD	*p*-Value
**Demographics**				
Male gender, *n* (%)	22 (63)	24 (69)	0.12	0.80
Age, years	47 (34–66)	44 (35–59)	0.24	0.32
**Comorbidities**				
Arterial Hypertension, *n* (%)	7 (20)	7 (20)	0.00	1
Diabetes mellitus, *n* (%)	0	2 (6)	0.35	0.49
Heart disease, *n* (%)	2 (6)	2 (6)	0.00	1.0
Previous neurological disease, *n* (%)	0	0	-	-
Alcohol, *n* (%)	10 (29)	10 (29)	0.00	1.0
Smoking, *n* (%)	0	0	-	-
COPD, *n* (%)	0	0	-	-
Liver Cirrhosis, *n* (%)	1 (3)	1 (3)	0.00	1.0
Cancer, *n* (%)	1 (3)	1 (3)	0.00	1.0
Chronic kidney disease, *n* (%)	1 (3)	0	0.24	0.99
**On ICU admission**				
APACHE score	18 (17–21)	20 (17–23)	−0.35	0.32
GCS score	4 (3–7)	10 (3–14)	−0.07	0.51
Hemoglobin, g/dL	12.1 (10.3–13.4)	12.2 (10.3–14.5)	−0.05	0.67
Glucose, mg/dL	129 (115–156)	131 (122–167)	−0.34	0.31
Sodium, mmol/L	138 (137–141)	137 (135–140)	0.38	0.26
Reacting pupils, *n* (%)	26 (74)	26 (74)	0	0.99
Traumatic SAH, *n* (%)	15 (43)	16 (46)	0.44	0.15
Epidural hematoma, *n* (%)	15 (43)	30 (86)	1.00	0.001
Hypotension, *n* (%)	15 (43)	16 (46)	0.05	0.99
Hypoxemia, *n* (%)	21 (60)	21 (60)	0.34	1.0
Marshall CT score			0.53	0.34
1	0	0		
2	1 (3)	2 (6)		
3	1 (3)	1 (3)		
4	7 (20)	2 (6)		
5	25 (71)	30 (86)		
6	1 (3)	0		
**During ICU stay**				
Mechanical ventilation, *n* (%)	35 (100)	35 (100)	-	-
Vasopressors, *n* (%)	26 (74)	33 (94)	0.57	0.05
Inotropic agents, *n* (%)	2 (6)	6 (17)	0.37	0.26
RRT, *n* (%)	0	1 (3)	0.24	0.99
ECMO, *n* (%)	0	0	-	-
EVD placement, *n* (%)	34 (97)	27 (77)	0.63	0.03
Intracranial Hypertension, *n* (%)	27 (77)	16 (46)	0.68	0.01
Seizures, *n* (%)	10 (29)	8 (23)	0.13	0.79
TIL score			0.65	0.09
1	5 (14)	1 (34)		
2	13 (37)	7 (20)		
3	2 (6)	5(14)		
4	15(43)	11(31)		
Brain tissue hypoxia, *n* (%)	-	24 (69)	-	-
Osmotic therapy, *n* (%)	21 (60)	18 (51)	0.17	0.63
Barbiturates, *n* (%)	10 (29)	4 (11)	0.43	0.13
Hypothermia, *n* (%)	8 (23)	10 (29)	0.13	0.79
Decompressive craniectomy, *n* (%)	11 (31)	11 (31)	0	1.0
**Outcomes**				
ICU length of stay, days	7 (3–14)	16 (9–25)	−0.77	0.001
Hospital length of stay, days	14 (3–41)	30 (14–66)	−0.39	0.03
GCS at ICU discharge	3 (3–12)	10 (3–14)	−0.20	0.46
Deaths at the ICU, *n* (%)	18 (51)	13 (37)	0.29	0.34
Deaths at the hospital, (%)	18 (51)	13 (37)	0.29	0.34
3-month GOS	1 (1–4)	3 (1–4)	−0.37	0.35
3-month UO, *n* (%)	25 (71)	19 (54)	0.36	0.22

COPD: chronic obstructive pulmonary disease; APACHE: acute physiology and chronic health examination; SOFA: sequential organ failure assessment; GCS: Glasgow coma scale; GOS: Glasgow outcome scale; ICU: intensive care unit; LOS: length of stay; TIL: therapy intensity level; RRT: renal replacement therapy; EVD: external ventricular drainage; SAH: subarachnoid hemorrhage; UO: unfavorable outcome; ECMO: extracorporeal membrane oxygenation.

**Table 3 brainsci-12-00887-t003:** Univariable and multivariable analysis of factors associated with the development of intracranial hypertension during ICU stay. Data is expressed as odds ratio (OR) and 95% confidence intervals (CI).

	Univariable Analysis OR (95% CI)	Multivariable Analysis OR (95% CI)
**Marshall score**	1.37 (0.83–2.26)	1.54 (0.84–2.82)
**GCS on admission**	0.96 (0.86–1.07)	1.00 (0.88–1.14)
**Reactive pupils**	0.30 (0.09–0.95)	0.28 (0.08–1.06)
**PbtO_2_ monitoring**	0.21 (0.09–0.50)	0.16 (0.06–0.42)

## Data Availability

Due to ethical reasons, datasets analyzed/generated in this study can be obtain through the corresponding author upon reasonable request. All other data can be found in the manuscript and in its Appendix A.

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
