# Peer review of "Brain Tissue Oxygenation-Guided Therapy and Outcome in Traumatic Brain Injury: A Single-Center Matched Cohort Study"

_brainsci, 2022, doi:10.3390/brainsci12070887_

Round 1
Reviewer 1 Report
Brain Sciences: Brain Tissue Oxygenation-Guided Therapy and Outcome in Traumatic Brain Injury: a single-center matched cohort study
This study used retrospective analysis to compare the effectiveness of intracranial pressure (ICP) guided therapy and ICP/PbtO2 guided therapy in severe traumatic brain injury patients. The authors provided statistical analysis from 35 matched patients from both groups. They did not find significant differences in hospital mortality or the occurrence of unfavorable neurological outcomes at 3 months between the two matched groups. However, patients treated with ICP-PbtO2 developed less intracranial hypertension than those monitored with ICP alone. They confirmed their finding in the entire cohort analysis (71 and 35 patients from ICP and ICP/PbtO2, respectively), however the authors concluded that results from future larger cohorts could be used to confirm their current study. Demographic data, clinical severity scoring, lab testing, and treatment were well recorded. The statistical methods used are appropriate for the study. However, there are some concerns that need to be addressed.
1. The treatment/monitoring assignment (ICP or ICP/PbtO2) was based on the availability of the device (PbtO2 probes), which became standard of care in their site since February 2017. This is respectful and reasonable. However, the authors should provide comparison between ICP and ICP/PbtO2 groups after Feb, 2017 as well. Although, there may not be enough population for the analysis power, it will provide information for future meta-analysis study.
2. On ICU admission, the ICP/PbtO2 group has higher incidence of epidural hematoma. The authors should provide insight information and discuss about the fact.
3. It appeared that ICP/PbtO2 group has longer ICU/Hospital length of stay. The authors should provide explanation and discuss about this fact.
Author Response
Reviewer 1
- This study used retrospective analysis to compare the effectiveness of intracranial pressure (ICP) guided therapy and ICP/PbtO2 guided therapy in severe traumatic brain injury patients. The authors provided statistical analysis from 35 matched patients from both groups. They did not find significant differences in hospital mortality or the occurrence of unfavorable neurological outcomes at 3 months between the two matched groups. However, patients treated with ICP-PbtO2 developed less intracranial hypertension than those monitored with ICP alone. They confirmed their finding in the entire cohort analysis (71 and 35 patients from ICP and ICP/PbtO2, respectively), however the authors concluded that results from future larger cohorts could be used to confirm their current study. Demographic data, clinical severity scoring, lab testing, and treatment were well recorded. The statistical methods used are appropriate for the study. However, there are some concerns that need to be addressed.
Authors’ response: Thanks for the nice summary.
- The treatment/monitoring assignment (ICP or ICP/PbtO2) was based on the availability of the device (PbtO2 probes), which became standard of care in their site since February 2017. This is respectful and reasonable. However, the authors should provide comparison between ICP and ICP/PbtO2 groups after Feb, 2017 as well. Although, there may not be enough population for the analysis power, it will provide information for future meta-analysis study.
Authors response: Thank you for your suggestion. We have added this information as a Supplemental Table. The mortality rate and the GOS at 3 months remained not statistically different between the two groups and the incidence of intracranial hypertension remained lower in the PbtO2-ICP group when compared to the ICP group. We have briefly discussed these results into the text.
- On ICU admission, the ICP/PbtO2 group has higher incidence of epidural hematoma. The authors should provide insight information and discuss about the fact.
Author’s comments: Thank you for your observation. Despite this discrepancy, in an adjusted multivariable model (age, PbtO2 monitoring, GCS on admission), epidural hematoma was not associated with mortality (0.43 C 95% 0.16-1.19) nor neurological outcome (0.70 CI 95% 0.23-2.10). As such, we have not reported further into the Results section.
- It appeared that ICP/PbtO2 group has longer ICU/Hospital length of stay. The authors should provide explanation and discuss about this fact.
Author’s comments: Thank you for your suggestions. We have added the following paragraph in the discussion section and we have also added a Supplementary Figure 2 (Kaplan Meier survival curve) to justify our comment : “Interestingly, patients monitored with ICP/PbtO2 presented longer ICU and hospital stays compared to the ICP group. This may be explained because patients in the PbtO2/ICP group, despite similar severity on admission, received more life support, which also prolonged survival time”.

Reviewer 2 Report
The authors present a manuscript describing a study that examines brain tissue oxygenation (PbtO2) guided therapy in a single center cohort. A retrospective analysis was performed on patients with severe TBI who received either ICP guided therapy (Jan 2012-Feb 2016) or ICP/PbtO2 therapy (Feb 2017-Dec 2019). Patients were genetically matched based on co-variates (demographics, co-morbidities, and injury severity scores on admission). The study background is done well and the methods appear to be well conceived and written. A few points about the study -
Results:
-Although it is stated that the ICP/PbtO2 was done at a later time point due to device availability and standard of care, how does the later time contribute to the findings- since they were not done during similar timepoints?
-Alcohol use was a considerable co-morbidity in 34 % of the patients, with 37% in the ICP group and 29% in the ICP/PbtO group. It would be helpful if you can explain either in the results or discussion how this comorbidity may have impacted the study- Initial GSC?
-It would be helpful if the discussion/conclucion offers points about the initial aims- understanding development of intracranial pressure, neurological outcomes and what this implies for global management.
Author Response
- The authors present a manuscript describing a study that examines brain tissue oxygenation (PbtO2) guided therapy in a single center cohort. A retrospective analysis was performed on patients with severe TBI who received either ICP guided therapy (Jan 2012-Feb 2016) or ICP/PbtO2therapy (Feb 2017-Dec 2019). Patients were genetically matched based on co-variates (demographics, co-morbidities, and injury severity scores on admission). The study background is done well and the methods appear to be well conceived and written.
Authors’ response: Thanks for the nice comments.
- Results: Although it is stated that the ICP/PbtO2was done at a later time point due to device availability and standard of care, how does the later time contribute to the findings- since they were not done during similar timepoints?
Authors’ response: We agree with the reviewer that time may have introduced a bias. As suggested by the other reviewer we performed a sensitivity analysis of ICP only monitored patients admitted during the period in which PbtO2 was available (Feb 2017 to December 2019) and we found similar results to the previous analysis, as shown in the new supplemental table S3. We have also added a supplemental Figure 1 showing the number of patients monitored in each group over time and the mortality rate and rate of unfavorable outcome per group per year.
- Alcohol use was a considerable co-morbidity in 34 % of the patients, with 37% in the ICP group and 29% in the ICP/PbtO group. It would be helpful if you can explain either in the results or discussion how this comorbidity may have impacted the study- Initial GSC?
Author’s response: Thank you for your comments. Alcohol exposure before a traumatic brain injury has been associated with better neurological outcome than in similar patients unexposed to alcohol. We think this issue is interesting but also difficult to discuss further in a limited cohort. It would be a nice topic to better evaluate in larger patients’ populations.
- It would be helpful if the discussion/conclusion offers points about the initial aims- understanding development of intracranial pressure, neurological outcomes and what this implies for global management
Author’s response: Thank you for your suggestions. We have added a paragraph discussing the management of intracranial hypertension as follows” Intracranial hypertension occurs frequently after TBI and is an important mechanism of brain injury that if left untreated can lead to death and poor outcome [34]. The Brain trauma foundation recommends starting treatment for intracranial hypertension when ICP reaches the value of 22 mmHg [5]. However, due to the increased risk of intracranial hypertension and its deleterious effects, the management of TBI patients should include strategies for prevention such as sedation and analgesia, elevation of the head of bed, optimizing of head venous return, regardless of the ICP level [28]. Moreover, a target of 20 - 22 mmHg may not be adequate for all patients. The use of adjunctive monitoring such as PbtO2 [35] can help individualized ICP target leading to further reduction of ICP in the presence of brain hypoxia or a higher tolerance for slightly higher ICP if PbtO2 is adequate and the patients is waking up.”
